# R.A.O. Project Recovery: Methods and Approaches for the Recovery of a Photographic Archive for the Creation of a Photogrammetric Survey of a Site Unreachable over Time

Vittorio Lauro [1],*, Marco Giovannangelo [2], Mariella De Riggi [3], Nicola Lanzaro [2] and Vittorio Murtas [1]

1   Department of Computer Science, University of Turin, 10124 Turin, Italy; vittorio.murtas@unito.it
2   Department of Ancient Sciences, University of Rome La Sapienza, 00185 Rome, Italy; lanzaro.1994719@studenti.uniroma1.it (N.L.)
3   Department of Cultural Heritage Sciences, University of Salerno, 84084 Fisciano, Italy; m.deriggi5@studenti.unisa.it
*   Correspondence: vittorio.lauro@unito.it

**Abstract:** The goal of this research is to make photogrammetric surveys of the walls of Cortona from 2012 accessible using new methodologies for recovering photographic material. This will allow a team of archaeologists to carry out a virtual reconnaissance of the surveyed stretch of wall as well as provide the basis for future investigations into any potential changes that may have occurred in the wall since 2012. Photogrammetry is a widely used technique in archaeology that can help researchers accurately measure, reconstruct, and analyze different architectural components of the wall. By using state-of-the-art photogrammetric techniques, including advanced computer vision algorithms, our team aims to produce high-quality 3D models and accurate measurements of different parts of the wall. The results of this research project will enable archaeologists to gain a more comprehensive understanding of the layout of the fortifications and the role of the Cortonese walls in the historical context of the area. Additionally, the research project will provide a detailed documentation of the wall that will be useful for both archaeological researchers and cultural heritage organizations. Finally, the research project will also provide the basis for future investigations into potential changes that may have occurred in the wall since 2012, which will be important for monitoring conservation and restoration efforts and providing an up-to-date record of the wall's state of preservation.

**Keywords:** photogrammetry; archaeology; cultural heritage

## 1. Introduction

The Recovery Cortona project, launched by the R.A.O., aims to analyze an archive of photographs taken in 2012 of the Cortona walls to identify the Etruscan and medieval phases of the walls. After receiving authorization from the Soprintendenza, modern photogrammetric techniques were employed in 2022 to develop 3D renderings of the walls. Specialists from various fields, including archeologists, photogrammetry experts, and computer scientists, collaborated to enhance the quality and depth of the gathered information.

The project faced the challenge of researchers being distributed across different locations, with some not having physically seen the walls. To overcome this, three-dimensional surveys were integrated into an interactive virtual model, allowing for remote collaboration and enabling archaeologists to conduct a reconnaissance of the walls and analyze them as they appeared in 2012. This approach allowed the project to leverage the potential of cutting-edge technology while respecting interdisciplinary and multidisciplinary parameters.

The virtual model created by the project facilitated a more accurate and detailed analysis of the Cortona walls, providing new insights into their history and evolution.

## 2. Methodology: Interdisciplinarity and Multidisciplinarity

The original archive of photographs used for the survey of the Walls of Cortona consists of 256 photos taken with a Panasonic DMC-Tz25 (image resolution 3000 × 4000) which were distributed in 8 folders divided by geographical location. The first activity was focused on the reorganization of the digital material, and the BeAPG protocol was chosen for this purpose [1]. The BeAPG protocol, born within the BeArchaeo project, is a protocol for the acquisition, processing, and export of photogrammetry data based on the principles of Digital Data Curation. It is characterized by not being based on specific hardware or software tools but rather by moving at a higher level of abstraction, adapting to the needs of each project. Following the BeAPG protocol, the photographic material was therefore reorganized according to chronological parameters (rearranging the photos along a collection arc from October 2012 to February 2013), geographical parameters (dividing the acquisition areas), and typological parameters (based on whether the object of the survey was a wall, a door, or a corner). In parallel, the team of archaeologists conducted an in-depth bibliographic research to catalogue all the information regarding ancient sources on the city of Cortona as well as the archaeological excavations carried out in the past on the city walls. At this point, the 3D models obtained from the BeAPG protocol were aligned, georeferenced, and inserted into the Unity 3D editor (version 2021.3.3f1) to create an interactive model that was sent to the archaeologists, who were able to virtually survey the walls of Cortona, studying and observing them as they were in 2012. The BeAPG protocol involves identifying the content of each photograph, assigning metadata to each image file, and organizing archives in a searchable manner. By using this protocol, archives can be expanded organically, with new surveys being seamlessly integrated into the existing archive without disrupting the workflow. Regarding the technique employed to process the photos, leveraging knowledge of the lens and camera model, as well as utilizing the lens distortion correction tools available in regard, the survey was conducted using the CMSV algorithm (Camera Multi-Stereo View). It was assumed that the photos had already been acquired with the purpose of photogrammetric surveying. The true challenge in such contexts lies not so much in addressing focus issues or inherent photo-specific intricacies but rather in organizing the source material in a way that it is managed organically and establishes a relationship among the acquired photographs, forming a cohesive survey maneuver. This maneuver encompasses the necessary movements that provide the machine with sufficient information regarding color point distribution and stereoscopic ratios, enabling the generation of the survey. For this reason, the Bea-PG methodology proved particularly valuable as it facilitated an organic data management protocol, resulting in the acquisition of the required surveys in less than two months.

A fundamental aspect of our research has been the adoption of an interdisciplinary approach, combining the expertise of scholars from different fields including computer science, medieval archaeology, and Etruscology. The aim of the research was not merely to adopt a multidisciplinary approach but rather to fully merge the contributions of each disciplinary perspective in order to create a new method of analysis capable of facilitating a broader overview and raising new questions for further research.

## 3. Literature in Reference to the Use of Photogrammetry in Unreachable Contexts

Researchers in the cultural heritage field are exploring the potential of reusing old photographic archives, which are no longer accessible or may depict a different state of the site, in new photogrammetric processing techniques. This is because traditional photogrammetric methods require images to be captured at the same moment in time, making it difficult to construct an accurate 3D model of cultural heritage sites that have undergone significant changes. Utilizing archived photographs allows researchers to capture the original state of the site and create 3D models that can account for changes over time. The use of old images can also help capture areas that are no longer accessible or exhibit different characteristics. However, there are challenges such as properly digitizing the images and correcting any distortions or occlusions caused by the camera model used to

capture them [2]. Photogrammetric recovery allows for the creation of highly accurate and detailed 3D models of cultural heritage assets, which can aid in restoration, conservation, and interpretation efforts. This approach overcomes limitations of traditional 2D imaging techniques, which lack depth information and cannot capture complex geometries and details. However, the quality, resolution, and format of images can impact photogrammetric processing, and proper digitization and pre-processing are essential for successful reconstruction [3].

Three-dimensional (3D) models created from archived photographs can aid in remote analysis and visualization, reducing the need for invasive physical interventions. Additionally, in cases of asset destruction, a digital copy must be created, as seen in the case of the Buddhas of Bamiyan [4]. In fact, three-dimensional imaging technology has become a significant tool in the preservation and restoration of cultural heritage assets. It has been used to create digital copies of deteriorating cultural assets to alleviate their destruction, and 3D-printing technology has expanded opportunities for cultural heritage preservation [5].

These replicas serve conservation and exhibition purposes as well as enable an exploration of culturally significant sites that may no longer be physically accessible. By allowing scholars from various disciplines to further research on such sites, 3D models can provide insights into the past, embodying the traditions, customs, and beliefs of a society, and prevent the loss of cultural heritage [6].

*Unreachable Archaeological Sites and Virtual Reality*

Many archaeological sites have been destroyed or altered over time due to natural disasters, human activity, or neglect. Some archaeological sites may be located in remote or dangerous locations, such as underwater or in war-torn regions. In addition, there are many sites that are too fragile to allow physical access, such as those in danger of collapse or those with fragile artifacts.

In such cases, where the archaeological site is unreachable, Virtual Reality (VR) can provide a safe and accessible way for researchers to explore and study these locations. In particular, using data from various sources, including archaeological surveys and excavation reports, researchers can create highly detailed and accurate 3D models of sites that have been damaged, altered over time or situated in hazardous places [7]. These models can be used for the analysis, visualization, and interpretation of the site, providing researchers with a more comprehensive understanding of the site and its features.

By creating immersive virtual environments that accurately replicate unreachable archaeological sites, researchers can explore, analyze, and interpret these places without physically visiting them. For example, VR was used to study the inaccessible Pleito cave, which is situated in the San Emigdio Mountains of California and is considered to be one of the most intricately decorated rock-art sites worldwide. By creating a VR interactive recreation of the site, researchers were able to explore and analyze the cave in ways that were not possible with traditional archaeological methods. In this virtual context, researchers can study and analyze the site without causing damage or degradation to the physical site itself.

Reconstructing through VR an unreachable location also has the potential to aid in site preservation. By creating digital archives of archaeological sites, researchers can document the condition of the site at a specific point in time, providing a valuable resource for future research and site management. One notable example is the digital reconstruction of the ancient city of Palmyra, which was destroyed by ISIS in 2015 [8].

Through VR, archaeologists and the public can now experience and study the city in its original state, aiding in the preservation and understanding of its cultural significance.

In addition, VR can be used to simulate the effects of different conservation and restoration approaches of unreachable sites, allowing researchers to test these approaches before (if possible) implementing them on the actual sites [9]. In the same way, VR technology

can be also used to simulate archaeological excavations [10], allowing researchers to test hypotheses and explore different scenarios in a controlled environment.

Our project too, presented in this article, concerns the virtual reconstruction of an unreachable site because the 3D reconstruction of the walls of Cortona was carried out starting from photographs taken in 2012, representing and preserving a unique and temporarily unreachable state of the archaeological site. Furthermore, thanks to VR, the archaeologists who collaborated on the project, despite not physically reaching Cortona, were able to observe and deepen the archaeological study of the Etruscan walls.

## 4. The Etruscan Walls of Cortona, Archaeological Context

### 4.1. The Etruscan Phase of the Walls of Cortona

The city of Cortona, located on a hill above the Chiana river, was founded in ancient times by the Etruscans. Despite having less funerary evidence than other Etruscan cities [11], Cortona's defensive walls remain intact and were later used as support for medieval constructions. The building of these walls was a ritual and symbolic act, representing the founding of a city [12]. In the Etruscan tradition, a well-founded city required at least three temples, roads, and gates [13]. The founding of Rome is a famous example [14,15], where Etruscan haruspices were summoned to implement the necessary rules for the city's correct formation [16,17]. While there is no known evidence of fortifications in Cortona prior to the existing walls, it is possible that perishable material constructions existed before their monumentalization [18]. However, the effects of natural and artificial taphonomic processes make it difficult to verify this hypothesis.

The study of city walls is a challenging task due to the complexity of the urban environment and the continuity of human activities. Despite this, some scholars have attempted to investigate the ancient city of Cortona through literary sources. One such source is the "New description of the ancient city of Cortona" by Domenico Tartaglini [19,20], which weaves together various literary sources and the biblical story to suggest that Cortona was one of the first urban settlements in Italy, which was established after the flood. Tartaglini also describes the grandeur of the city walls, which he believes were built by giants. Paolo Uccelli's "History of Cortona" [21] also highlights the strength and richness of the city, as demonstrated by the strong walls and grandiose ruins found both inside and outside the boundary wall. However, Uccelli also notes the challenges of researching the city's history due to the destruction of contemporary public documents. Despite these challenges, the study of tangible data remains a convenient but less safe approach to understanding the history of ancient cities.

The first archaeological excavations of Cortona were directed by Aldo Neppi Modona in 1929 [22,23]. Neppi Modona's excavations were commissioned by Antonio Minto, the then R. Superintendent of Etruria, and they were carried out to compile quadrant IV of the archaeological map of Italy. The excavations focused on various points of the wall layout and revealed the Etruscan layout of the defensive walls. The presence of waste water drainage channels in conjunction with the gates on the SW and NW sides was also reported. Neppi Modona noted that the walls were founded directly on the local sandstone, specially leveled for installation, and ascertained the existence of a second row worked below street level, which demonstrates a subsequent raising of the walking surface. The blocks appear regularly arranged on the external face, while internally, they are irregular to form random protrusions, and some blocks of considerable size cover the entire width of the wall. The portion of the wall near Porta Colonia in the direction of Porta S. Maria shows an expedient to stem the changes in altitude of the land [24,25]. In total, a perimeter of 2.880 m and an area enclosed by the walls of 35.57 ha is estimated. Additionally, avant-corps external to the walls are reported, perhaps towers or having the function of protection. Neppi Modona hypothesizes a further door near the polygon (Figure 1).

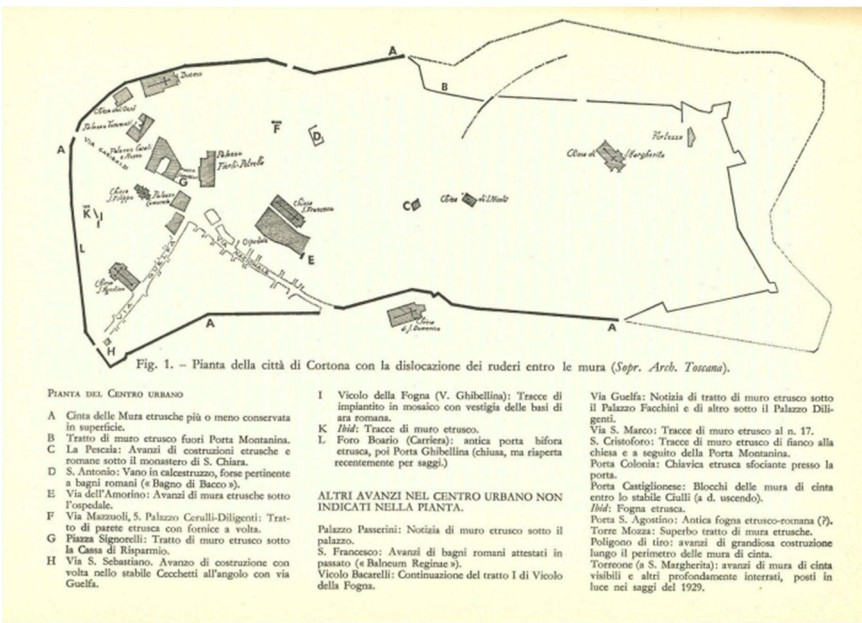

**Figure 1.** Representation of Cortona as reported by Aldo Neppi Modona, Cortona: Etruscan and Roman history and art.

The ancient city of Cortona has several gates, with only one, the Bifora or Ghibellina gate, being definitively placed in the Etruscan era [25]. The other gates are believed to be medieval, although traces of a closed door can be seen between Porta S. Maria and Porta Colonia. Porta Sant'Agostino was moved and raised in the 19th century. Near the Bifora gate, two bronze statuettes of Culsans and Selvans were discovered, suggesting the presence of a sacellum [26]. The gates held symbolic value in the ancient world, with the arched gate serving as a passage between the living and the dead or the city and the forest [27]. Recent excavations from 1987 to 1995 revealed a single-span door system from the end of the 4th century B.C. or shortly before the end of the 3rd century B.C., which was later buried to construct a double arched door [28]. The original arch and central pier are still visible, and there may have been a tower leaning against the external face. However, the complex history of the gate's construction has made it difficult to precisely date or identify the building materials used [29]. Therefore, modern technologies and methods should be utilized to gain a more accurate and comprehensive understanding of the defensive circuit of Cortona beyond just specific sections of the walls.

*4.2. The Medieval Phase of the Walls of Cortona*

The news regarding early Christian and early medieval Cortona is scarce: Paolo Uccelli, in his History of Cortona, after having recounted the ancient history of the city, simply reports that "[. . . ] the few memories that remain of Cortona from that time on are authentic that his condition gradually deteriorated [. . . ]" [21] and that, although it was probably one of the seats of the main Magistrate of Tuscany, it did not have a particular political relevance in the territory of central Italy, at least in this era. Furthermore, the invasions, first Gote and then Longobards, caused a further weakening of the city from a political and commercial point of view, until, with the advent of Charlemagne in Italy, the city experienced a period of recovery.

However, it was after the year 1000 that Cortona became a more powerful center, above all commercial, thanks above all to its strategic position which, between the plain and the Apennines, acts as a point of connection for trade between the plain delle Chiane and the Valtiberina, as also testified by the documents of the *Registrum Vetus* [30].

In the thirteenth century, from a political point of view, Cortona became a free municipality, forging alliances with the cities of central Italy, during the troubled years of

the clashes between Guelphs and Ghibellines, which saw it in opposition with the city of Arezzo, so much so that in 1258, it was sacked and occupied and to be resumed only three years later [21].

On the occasion of the birth of the municipality of Cortona, the building activity of the city experienced a moment of intense activity, which sees, first of all, the restructuring of the road axes that connected the Ghibellina gate with the Colonia gate and the San Domenico gate together with the construction of buildings that are symbols of city power, such as the Palazzo Comunale, the Pieve di Santa Maria and the fortress of Girifalco [21].

As regards specifically the city walls, these widened, in some points, especially in the eastern part of the city, incorporating the external villages which, starting from the 11th century, began to form outside the main core of the city itself: it is, in fact, a wall that starts from the current church of the Holy Spirit and which continued up to the mullioned door [25], thus incorporating the village which, still today, takes the name of the village of San Vincenzo. A similar thing happens for the village of San Domenico, equipped with external doors and a wall, equipped with cylindrical towers, which surrounds it as early as the thirteenth century [21].

As already mentioned, in the thirteenth century, Cortona was at the center of political events, which led it to face and suffer a siege and, after the recovery of their city by the Cortonese, the restoration of the city walls was essential: in part, the reconstruction followed the previous Etruscan layout of the walls, but the most intense reconstruction works were concentrated in the points where the Fortress of Girifalco connected with the access points to the city. The gates, in fact, were also rebuilt in the following centuries, so much so that in the 16th century, bastions were added in front of the city gates, which are currently no longer visible because they have been demolished, with the exception of the one next to the holy gate, as shown in Figure 2 [21].

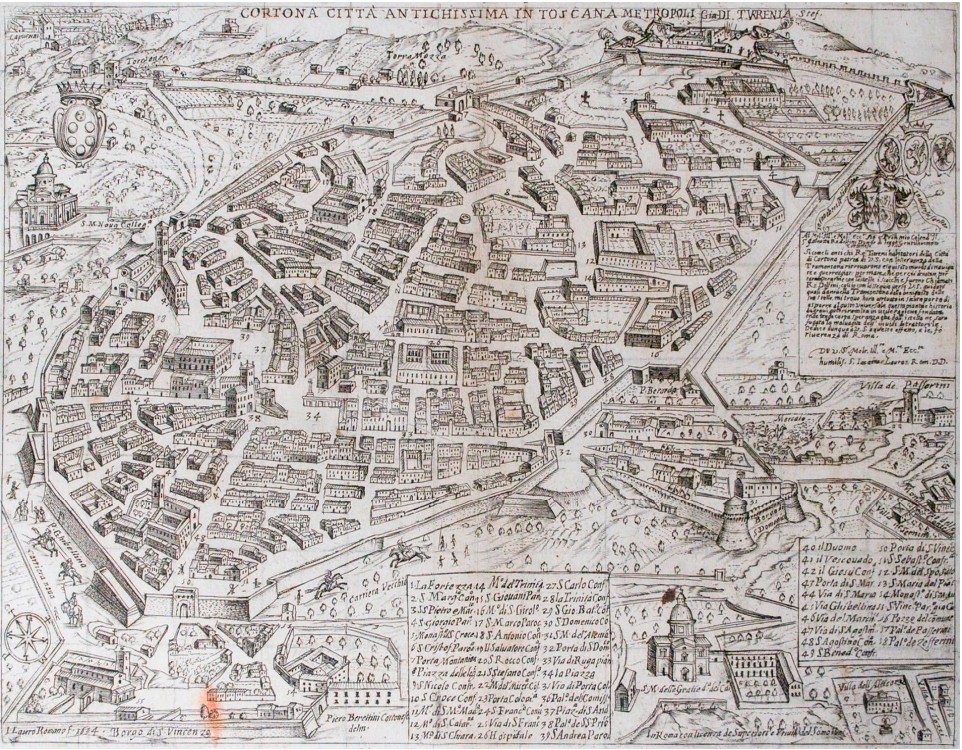

**Figure 2.** Illustration of the city by Pietro da Cortona cit. in bibliography.

## 5. Technical Insight

### 5.1. Analysis of Satellite Images and GPS Positioning

After a thorough examination of the studies on the walls of Cortona, both during the Etruscan and medieval phases, we now return to the three-dimensional model. As can be observed, this model has not been georeferenced solely based on the textual sources that attest to the position of the various elements but also on satellite images. This topographical study was essential in creating the interactive virtual experience, which subsequently allowed the archaeologists of the team to conduct a virtual reconnaissance in situ and carry out an initial investigation of the territory without ever physically visiting the site.

This approach, which merges traditional archaeological methods with cutting-edge technology, has opened new avenues of research.

In order to produce a systematic analysis, we produced a geodatabase to interpolate the archaeological dataset and data available from a primary topographic analysis. We use the software QField v.3.28.4-Firenze; the chosen is related to its accessibility because the software is open source. We also chose the version 3.28.4-Firenze because it is the most equilibrated version in term of stability and downloadable tools or plugin.

The chosen SRS is RDN2008/Italy zone (EPSG: 7794), the newest Italian RS in effect since 2008, which has the advantage of covering the entire nation [31]. The more used RS such as Monte Mario/Italy zone 1 (EPSG: 26591) and Monte Mario/Italy zone 2 (EPSG: 26592) are based on the Monte Mario Meridian, which passes through the Cortona area. Therefore, the analyzed area llies on the border between EPSG: 26591 and EPSG: 26592.

The project operated on four separated parts of the city wall, as shown in Figure 3. This division in sector does not reflect a difference in masonry, periodization or building technique; this division is based on the continuity and discontinuity of the produced 3D model.

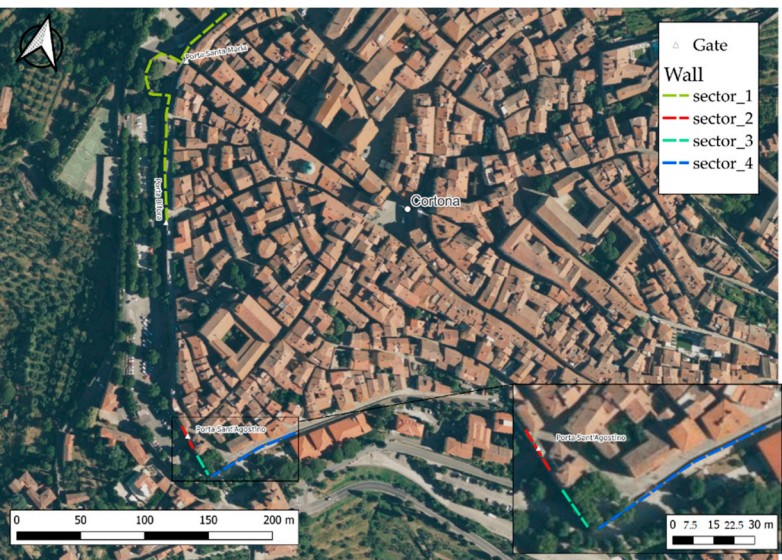

**Figure 3.** Map of Cortona with processed city-wall sectors and its city-gates; Basemap: © Bing Aerial, 2018.

We produce a comparison between the geomorphology and enclosure wall [32]. The contour lines are extracted by high-quality DEM with a resolution of 10 m cell size produced by INGV of Pisa and available for all the Italian territory. It is clear how the walls were built according to the tactical edge of the plateau, avoiding the area with steeper slope. The sectors interested by this research lay at an altitude between 460 and 470 m a. s. l. A solution to avoid the presence of a steep slope inside the city wall with the construction of low terraces has been already observed by Neppi in other parts of the city [33], but actually, this detail is not visible in our zone from the remote sensing analysis.

Actually, we were able to detect and map the Etruscan original block incorporated in the most recent sections of the enclose, as is shown in Figures 4 and 5.

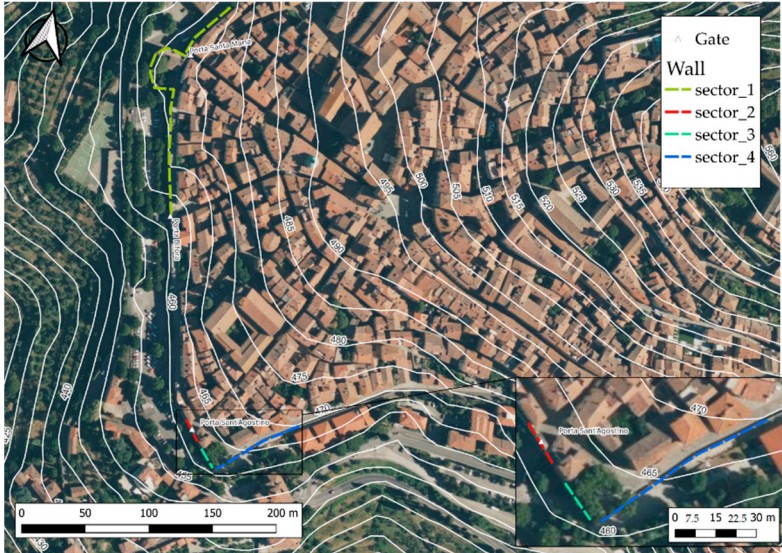

**Figure 4.** Map of Cortona with processed city-wall sectors, city-gates and contour lines; Basemap: © Bing Aerial, 2018.

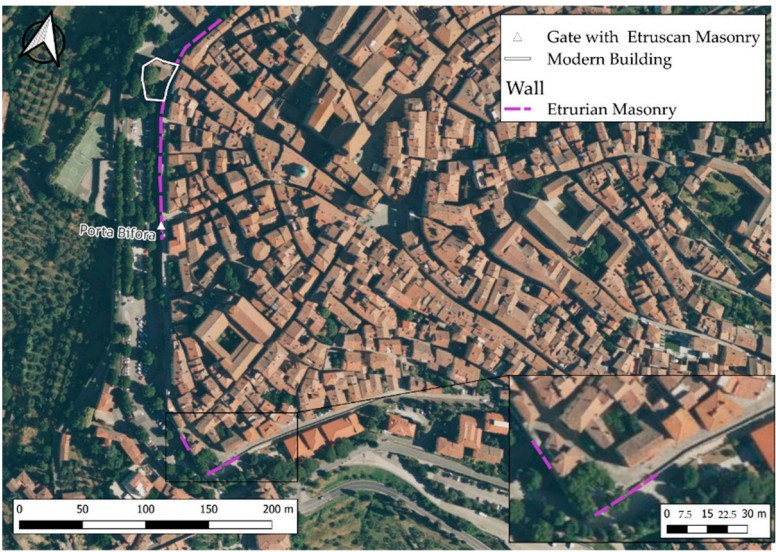

**Figure 5.** Map of Cortona and remains of Etruscan masonry present in the investigated area; Basemap: © Bing Aerial, 2018.

The spot of anomalies and architectural details will be our priority in the direct survey activities of the area in the program before the end of the year. After a complete mapping of these details, we could obtain a wider dataset useful for reconstructing the city-wall lifecycle. All data will be collected on a work field via QField; this is a software available for Android OS which allows uploading data from the field on an imported geodatabase previously built up on QGis. These data and the already collected ones will flow on a final geodatabase in order to be a starting point for an ultimate historical reconstruction.

### 5.2. A VR Application to Represent the Walls of Cortona

To provide support for the archaeological investigation, it was decided to create a non-immersive virtual reality application that could allow archaeologists to carry out a

reconnaissance process of the walls of Cortona. In this way, archaeologists could observe the virtual walls and share their analysis with other collaborators.

In particular, a desktop application (for Windows) was created using Unity 2021. The work pipeline involved cleaning up and aggregating the models obtained using photogrammetry techniques, exporting the models to the game engine, creating the virtual terrain, and finally building a first-person cursor to navigate the environment.

The first-person cursor was developed by taking advantage of Unity's new input system, ensuring that the application can be used both via a keyboard and joypad. The link between user-supplied input and the code that ensures cursor movement was achieved through the input system's messaging system and a ScriptableObject. In addition to horizontal movement and camera rotation, the cursor has a function of increasing the speed of movement and has the ability to jump and temporarily float in the air.

The virtual terrain was constructed starting from the 3D models' morphology: specifically, sections of the virtual environment were raised and made to coincide with the underside of the walls. The purpose was to recreate the real height of the city of Cortona and the vertical offset between the various streets. In the process of creating the virtual terrain, geometric details useful in the exploration of the walls were inserted. In particular, planes positioned transversely to the walls were added (Figure 6). These objects serve as platforms for viewing the upper parts of the findings and are accessible by the user using the cursor jump function.

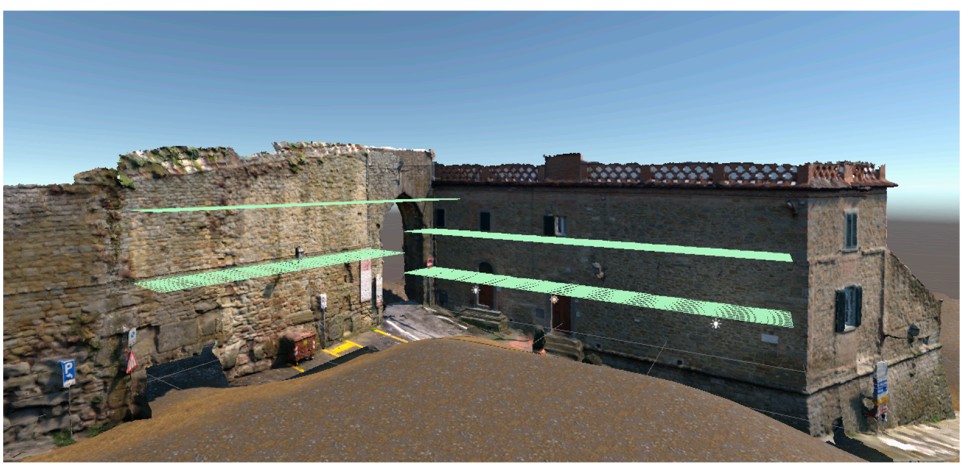

**Figure 6.** Virtual platforms to visit the upper parts of the walls.

If for the medieval era, it is possible to obtain some information thanks to historical records, this is definitely more difficult for earlier periods. Therefore, the creation of a 3D model that can allow remote recognition by scholars and specialists can make the phase preceding the study of the remains of the walls faster and more efficient, especially if this model can use accessory tools as metric and geographical references together with a greater definition of the images themselves and the possibility of also observing the internal face of the walls. However, it is important to underline the double potential that this model could have: if, in fact, on the one hand, some technical improvements could help archaeologists in the study of the walls and their phases, on the other, the functionality that this product could have in the context of dissemination and knowledge of the archaeological asset addressed to the general public. The "immersive reality", especially in recent years, is creating an increasingly large audience in favor of archaeological sites, museums and galleries. Without, therefore, distorting the tool which, in any case, was created for a scientific purpose, if adapted to the needs of a wider and less specialized public, it could also be transformed into a popular product, making the walls of Cortona an integral part of the experience by visiting the city and its monuments.

The 3D model developed with the integration of photographs makes it possible to map and study, with an adequate degree of definition and integrated tools, a mass of data,

inter alia in a cross-referenced way and with an incomparable speed, theoretically without going on site, providing a series of morphological and topographical details that it would be difficult to appreciate synoptically. On the other hand, the possibility of also using older photographs could insert details that would otherwise be lost [34,35].

## 6. Results

Our research is still ongoing. The BeAPG protocol allowed for the recovery, reorganization, and re-elaboration of data collected in 2012. The creation of an interactive three-dimensional model enabled researchers to collaborate over long distances. The computer scientists provided the humanists with an interaction with the virtual context that aligned with their methodological approach of reconnaissance, demonstrating the multidisciplinary and interdisciplinary potential of this method. The work is still ongoing, and final results are subject to the definitive verification by the Soprintendenza of archaeology, fine arts, and landscape for the provinces of Siena, Grosseto, and Arezzo. Nonetheless, the research's potential is fully expressed in the cooperation put into practice. Further analysis is still necessary to determine the model's accuracy and the research's overall effectiveness.

## 7. Conclusions

This intervention represents the first interdisciplinary approach to the walls of Cortona, which is aimed at conducting a comprehensive study of a site that is currently inaccessible. This approach provides an important foundation to compare the present state of the walls with that of 2012, identifying any critical elements or degradation. The objective of the R.A.O. is to concentrate on creating an interactive model equipped with accurate measuring tools or zoom for detailed analysis of the walls. Additionally, with the approval of the superintendent and after conducting more photogrammetric surveys, the goal is to create an online interactive model that allows anyone to view and visit the walls of Cortona virtually.

This interdisciplinary approach aims to gather data from multiple fields, such as geography, history, and archeology, to fully understand the origins of the walls of Cortona and their influence on the town's development. The team aims to use these data to create a comprehensive map of the walls, detailing their historical significance and influence on the town's fortifications. By doing so, the team hopes to not only preserve the walls' historical significance but also develop new opportunities for tourism and education.

The creation of an online interactive model of the walls of Cortona provides new research opportunities for scholars studying the region's history and architecture. By using photogrammetric surveys and historical records, the team aims to create an accurate representation of the walls that will be accessible to anyone worldwide. This would provide new opportunities for international collaboration on projects involving the history and culture of Cortona, further advancing interdisciplinary approaches to the study of this region.

**Author Contributions:** Conceptualization, V.L.; Methodology, V.M.; Software, V.M.; Investigation, M.G.; Writing—original draft, V.L., M.G., M.D.R., N.L. and V.M.; Writing—review & editing, V.L.; Supervision, V.L. All authors have read and agreed to the published version of the manuscript.

**Funding:** This research received no external funding.

**Acknowledgments:** This project was made by the R.A.O. and has not received any kind of external funding. The RAO project was conceived within the activities of the Morkedainen Art & Craft Department with the purpose of fostering research and development in the realm of cultural heritage by means of projects aimed at safeguarding, conserving, and disseminating the patrimony of cultural heritage in Italy and worldwide. The achievement of this goal is pursued through participation in calls for proposals and competitions for the realization of the project itself or by promoting new projects aimed at safeguarding, conserving, and disseminating the patrimony of cultural heritage. Thanks to the superintendence of archaeology, fine arts and landscape for the provinces of Siena Grosseto and Arezzo which authorized this activity with authorization #66248173 and to Ada Salvi, who is specifically responsible for the procedure.

**Conflicts of Interest:** The authors declare no conflict of interest.

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
