# Peer review of "R.A.O. Project Recovery: Methods and Approaches for the Recovery of a Photographic Archive for the Creation of a Photogrammetric Survey of a Site Unreachable over Time"

_heritage, doi:10.3390/heritage6060250_

Round 1
Reviewer 1 Report
The paper R.A.O. Project Recovery: methods and approaches for the recovery of a photographic archive for the creation of a photogrammetric survey of a site unreachable over time aims to present the results of a photogrammetric survey of the city walls of Cortona in ancient Etruria targeted at the conservation and divulgation of the accessible and inaccessible walls of the ancient town. Unfortunately, the paper does not provide useful elements for evaluating the methodology and the results of the project. Even if an appreciated introduction about the research branches of photogrammetry in current archaeology is presented, the paper is not adequately structured in paragraphs and sub-paragraphs. The methodology is not clearly stated, repetition is recurrent and many paragraphs could be substantially summarized. Even the ultimate purpose of the work is not clear. It seems to report a work in progress, presenting some of its highlights, but without really defining the overall target and the specific goals of the individual phases
In particular, the structure of the paper should be rearranged with: 1. introduction and state of art; 2. Context of research and target of analysis; 3. Materials and methods; 4. Results; 5. Discussion and summary. This is crucial to ensure that the reader does not lose himself within the manuscript. Unfortunately, none of these points are reported and the paper in its actual structure appears weakly structured and rather redundant in several parts.
Just a couple of suggestions:
· Point 4 should become: materials and methods
· Point 5 should be placed before point 4 and titled as: Context of research, corroborated with a brief definition of the target of analysis
· Point 6 satellite images should be put in a sub paragraph of materials and methods
The process for the realization of the virtual model should be described as the reference photographic datased clearly defined. The workflow is not explained nor are bibliographical references given where specific processing or parts of the project and analytical techniques adopted have been previously published, apart from reference [7], generally describing the analytical protocol adopted.
The link between the use of satellite images and DEM/DTM with the photogrammetric elaboration of on-land images is not clear, nor their cross-correlation.
The graphic apparatus should be substantially corroborated with more images, also reporting the processing workflow of the elaboration.
Lastly, reference literature in the introduction and in the other parts of the text should be substantially increased.
In conclusion, in my opinion the manuscript in its present for is not suitable for publication.
Author Response
Dear Reviewer,
Thank you for your valuable comments on our article "RAO Project Recovery: methods and approaches for the recovery of a photographic archive for the creation of a photogrammetric survey of a site that was unreachable over time". We have attempted to make the requested changes to improve the structure and clarity of the article.
Regarding the structure of the article, we followed his suggestion and reorganized the text into five main sections focusing on introduction and state of the art, research context and objective of the analysis, materials and methods, results, discussion and synthesis. We have also clarified the specific objectives of the individual phases.
We have also revised the process description for making the virtual model
We hope these changes are enough to improve our work and make it suitable for publication. Again, thank you for taking the time to review our article.
Best regards,
The team of authors.
Reviewer 2 Report
Major questions: Does this paper present actual results or is it a future proposal towards the 3D representation of the Cortona Walls?
Generally, I am not very sure about the technical readiness of the paper. The term “photogrammetry” is now used almost explicitly for the creation of 3D models from a large set of images and in this work, there is no clear evidence for this image dataset.
For the authors, it seems to me that the most appropriate way to describe their work, is to present it as an attempt to digitally reconstruct the walls of Cortona based on a set of old time photos and other additional sources of information. However, if this is the case, this is called 3D reconstruction and is based on a rather artistic pipeline of work, interdisciplinary research, which describe better to the actual work or proposal of the authors. When a researcher uses older photos that represent a past condition/state of a monument to create a 3D representation, this is not photogrammetry but 3D digital reconstruction of the monument (a kind of timeline let’s say).
Moreover, there are a lot of generalities in the paper like the following in the abstract (line 21-23):
“By using photogrammetry in combination with the latest technological advances, it is possible to recover photographic material that was previously not accessible, and to obtain new information about the structure of the wall, the nature of its maintenance, or the characteristics of the materials used to build it.” This is totally incomprehensible.
Another important issue that is addressed somehow optimistic is the fact that the authors seem to rely on photogrammetry to create high precision and accuracy 3D models. However, is it widely known that metrical accuracy is reliable when using terrestrial laser scanners and LiDAR, especially when large scale monuments are considered.
Minor questions:
What the acronym R.A.O. stands for?
Sections 3 and 4 have the same title. I suppose that the authors would like to present a discrete review over traditional photogrammetric methods and newer ones (photogrammetry for 3D representations). In any case this should be corrected.
Is the WebXR-based applications called Aton ready for demonstration? This is unclear in section 8. In any case, even if this application is not fully ready, some screenshots of the prototype would add significantly to the technical soundness of the paper.
Virtual platforms in Unity: While this is a good idea, the authors may use also a flight camera that simulates the movements of a drone.
By Superintendent (line 620), does the author mean the ministry of culture or department/ephorate of antiquities?
A minor editing of English is necessary.
A minor editing of English is necessary. Please check carefully, there are several syntactical errors throughout the manuscript.
Author Response
Dear Reviewer,
We thank you for your evaluation of our article and for providing us with some precious indications to improve it. We are pleased to hear that you believe our work has good potential and that the topic is of interest.
We took your recommendations into consideration and completely rearranged the text following the suggested structure.
In the Introduction section we have described the project, its purpose and the importance of the photogrammetric method and virtual reality in unreachable contexts and in wall analysis. We then added a State of the Art section, supported by proper references, which was missing in the previous version.
We then combined paragraphs 5 and 6 in the Archaeological context section using a more concise and orderly language and syntax.
In the Method section we have precisely described the experimental conditions of the adopted method, including the number of images considered to carry out the survey, the resolution and the protocol used.
In the Findings and discussions section we have rearranged paragraph 7, as suggested, and added other information currently present in other paragraphs.
Finally, we have merged paragraphs 8 and 9 in the Conclusions section.
We are confident that these changes have made the text clearer and more organized. We hope that the new version of our work will please you and satisfy your requests.
We thank you again for your valuable feedback.
Best regards,
Reviewer 3 Report
The presented manuscript illustrates the results of a photogrammetric survey carried out starting from archival photographs in order to study the Etruscan walls of Cortona and to produce a digital document useful both for remote study and for the "virtual" conservation of the site itself. For these reasons, the paper could be of great interest to the scientific community, however it has some gaps that should be filled.
Below are some comments. Please also see the attached pdf.
The abstract is too long and contains information that is widely discussed. It should be summarized indicating the topic, objectives, methods and summary of the results obtained.
The introduction contains some information (see pdf) that is redundant and present everywhere and in a confusing way in the text. They should be eliminated.
Paragraph 2 could be completely deleted.
It is mandatory the increasing of the discussed bibliography.
Paragraph 3 and 4 have the same title.
All sections 1-2-3-4 have many redundancies that should be eliminated.
There is no actual paragraph showing the results, but they are scattered throughout the text.
It would be necessary to insert some images (even screenshots) of the 3D survey.
Please, eliminate spaces within the same paragraph.
In general I believe that the manuscript has good potential and that the topic is of interest but I suggest completely reorganizing the text as the information, although present, is written in a confused and verbose way.
You could follow a structure like this:
1. Introduction: description of project, aims, importance of photogrammetric method and virtual reality in unreachable context and in masonry analysis (remote studies, multidisciplinarity and interdisciplinarity).
1.1 State of the art (photogrammetry and virtual reality, currently paragraph 7). It should be supported by adequate references, which are missing.
1.2 Archaeological context. Description of the archaeological context, combining paragraph 5 and 6 in a single paragraph. You would be able to report all the information equally but you should use a more concise and orderly language and syntax.
2. Method. Describe precisely the experimental conditions of the adopted method (number of images considered to carry out the survey, resolution,...) and the protocol.
3. Results and discussions. Rearrange the current par. 7 and add other information currently present in other paragraphs (for example: lines 210-223, 442-447, 454-494), adding one or more figures.
4. Conclusions. Merge paragraph 8 and 9.

The English is correct but the language is too verbose and not very synthetic. It should be revised to make the text more readable and understandable.
Author Response
Dear Reviewer,
First of all, we would like to thank you for your valuable comments regarding our article. We have taken every comment seriously and made the necessary changes to improve our work.
In relation to the abstract, we have shortened the description of the historical-archaeological context, focusing more on the objective of our study, the methodologies used and the main results obtained.
As for the introduction, we have eliminated redundant and confusing information and rephrased the text in a clearer and more incisive way. Also, we have completely eliminated paragraph 2, as suggested and now it is a completely transformed one.
We have changed the title of paragraph 4 to make it distinct from paragraph 3. We have also eliminated the redundancies present in all the sections, to make the text clearer and more flowing.
Finally, we have inserted some images of the 3D survey, including some screenshots, and we have created a special paragraph to show the main results of our research.
Again, we thank the reviewer for their valuable input and hope that the changes they have made have improved our article.
Best regards,
Round 2
Reviewer 1 Report
In my opinion the paper is fairly improved in his structure and clarity. The effort of the authors in increasing the readability of the edited draft is appreciable.
However, I still do not comprehend the overall target of the research and the final outcomes. I would like to see more about the adopted workflow for the cleaning and processing of new and old photos (archive documentation) to produce the 3D model. How did you managed to create the 3D point cloud processing old picture that probably were taken in different periods, with a different exposure to light and state of preservation of the structures of interest? This is an important point that should be further deepened to present the reader an effective and metrically correct digital model, produced based on “old” photographic documentation.
The written English should be also improved and edited by a proficient English proof-reader.
I suggest minor revisions.
In my opinion the paper is fairly improved in his structure and clarity. The effort of the authors in increasing the readability of the edited draft is appreciable.
However, the workflow adopted for the creation of the 3D point cloud, in particular in relation to old photographic documentation, should be described in more detail.
Written English should be increased too.
I suggest minor revisions.
Author Response
Dear Reviewer,
Thank you for your valuable feedback. In response to your comments, the manuscript has undergone a thorough revision. We have addressed your concerns by adding a section in paragraph 2 that delves into the technical aspects of the conducted surveys. However, the focus of our discussion primarily revolves around the organizational and protocol-related aspects of the survey, rather than emphasizing the instrumentation used (although it has been duly mentioned). The aim of our article is to showcase the potential of a multidisciplinary approach rather than engaging in a detailed exploration of the photogrammetric technique.
We sincerely appreciate your guidance, and we will proceed to upload the revised paper today.
Yours sincerely,
Reviewer 3 Report
From the file version, you don't see the changes you made, so it's really hard to evaluate the revision.
The article is still too confused in structure. Much information is redundant, there are paragraphs with the same title, the bibliography relating to photogrammetry has not been enriched at all. The results paragraph is non-existent.
Work should be reviewed properly by indicating changes and responding to all reviewer comments.
The English Language needs some minor editing
Author Response
Dear Reviewer,
I apologize for any confusion caused by my previous response. It appears that you may have reviewed the earlier version of the article before the revision. I assure you that we have thoroughly addressed your suggestions, and there are no longer paragraphs with the same title. The entire structure has undergone significant revisions based on your valuable feedback. We genuinely appreciate your guidance, which served as the foundation for a comprehensive revision that has substantially transformed the text.
Thank you once again for your insightful recommendations.
Sincerely